# The Surprising Effectiveness of Test-Time Training for Few-Shot Learning

Ekin Akyürek [1]   Mehul Damani [* 1]   Adam Zweiger [* 1]   Linlu Qiu [1]   Han Guo [1]   Jyothish Pari [1]
Yoon Kim [1]   Jacob Andreas [1]

## Abstract

Language models (LMs) have shown impressive performance on tasks within their training distribution, but often struggle with structurally novel tasks even when given a small number of in-context task examples. We investigate the effectiveness of test-time training (TTT)—temporarily updating model parameters during inference using a loss derived from in-context examples—as a mechanism for improving LMs' reasoning and few-shot learning capabilities. On the Abstraction and Reasoning Corpus (ARC), performing TTT with in-context examples yields up to $6\times$ higher accuracy compared to fine-tuned baselines—reaching 53.0% on the public validation set with an 8B-parameter LM and 61.9% when ensembled with program-synthesis methods, matching average human performance. On BIG-Bench Hard (BBH), TTT on in-context examples surpasses standard few-shot prompting in the 10-shot setting by 7.3 percentage points (50.5% to 57.8%). Our findings highlight the limitations of in-context learning for novel tasks and demonstrate the potential of test-time training to enhance language model adaptability.

## 1. Introduction

Large-scale neural language models (LMs) have demonstrated remarkable success on few-shot learning of tasks related to those seen during pre-training, as well as elementary variations or compositions of those tasks (Brown et al., 2020; Todd et al., 2024). When given natural language specifications or a small number of examples, LMs can often infer the desired task and generate appropriate outputs. However, an open question is whether these models can *truly* acquire new skills for which they have not been

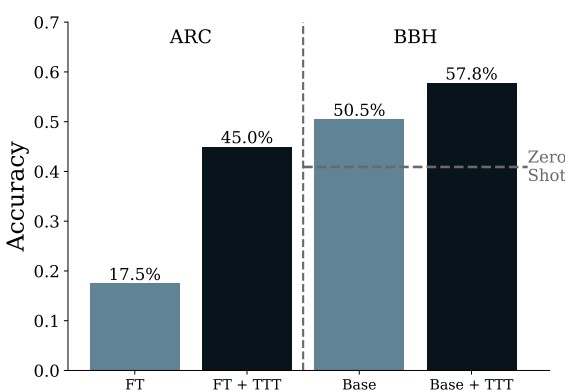

Figure 1. Pass@2 accuracy on a subset of 80 randomly selected ARC validation tasks and overall accuracy on BIG-Bench Hard. The zero-shot baseline is 0 for ARC and $40.9\%$ for BBH, indicated by the dashed line. TTT boosts the performance of fine-tuned models (FT) on ARC by 27.5 percentage points and increases accuracy on BBH by 7.3 percentage points.

trained—particularly, tasks involving non-trivial reasoning, planning, and abstraction in domains that differ significantly from their pre-training distributions. This question is fundamental to understanding how, and whether, LMs can exhibit the sort of flexible, novel-skill acquisition that has been proposed as a measure of intelligence (Chollet, 2019).

Solving *complex and novel* tasks remains extremely challenging for LMs, and simple sampling approaches often yield poor performance on such problems (Wu et al., 2024; McCoy et al., 2024). However, recent progress has shown that LMs can be substantially improved by adding extra *test-time computation*. Several methods fall into this category, such as chain-of-thought prompting (Wei et al., 2022), sampling with majority voting (self-consistency; Wang et al., 2023), code execution (Brown et al., 2024; Snell et al., 2025; Damani et al., 2025), and search (Yao et al., 2023).

The idea of updating model parameters using *instance-specific* data at test time has roots in the literature on *local learning* (Bottou & Vapnik, 1992) and *transductive learning* (Joachims, 1999). In these methods, a learner refines its parameters or hypotheses *after* observing test inputs, adapting to individual examples or small clusters of examples. Such approaches inherently blur the line between training and inference, and can lead to robust adaptation in low-data

---
*Equal contribution   [1]Massachusetts Institute of Technology. Correspondence to: Ekin Akyürek <akyurek@mit.edu>.

*Proceedings of the $42^{nd}$ International Conference on Machine Learning*, Vancouver, Canada. PMLR 267, 2025. Copyright 2025 by the author(s).

scenarios or under distribution shift.

Modern versions of these transductive ideas for deep neural networks have been widely referred to as *test-time training*. In TTT, a model is updated at inference time using only the current test instance or a small batch of test instances, typically through explicit gradient steps. While test-time adaptation has been explored for vision models (Sun et al., 2020) and sequence architectures (Gandelsman et al., 2022; Sun et al., 2024; Behrouz et al., 2025), its interaction with other techniques for few-shot learning—especially in-context learning—remains less understood.

In this paper, we investigate how to leverage TTT *on top of* standard in-context learning (ICL) to boost performance on challenging tasks that require reasoning or rule-based generalization. In-context learning is a powerful means of adaptation *without* parameter updates, guided by short, task-specific prompts. We show that combining ICL with explicit gradient-based updates on test data can significantly improve performance on particularly difficult tasks. Specifically, our main contributions[1] are:

1. A systematic analysis of the key components for effective test-time training, including strategies for selecting training data at inference, training objectives, and how TTT interacts with an LM's pre-trained parameters and in-context learning.

2. An application of TTT to two challenging benchmark suites—**The Abstraction and Reasoning Corpus** (ARC; Chollet, 2019) and **BIG-Bench Hard** (BBH; Srivastava et al., 2023; Suzgun et al., 2023).

On ARC, our TTT approach outperforms existing open-source neural methods, attaining 53.0% accuracy with an 8B model and 61.9% when ensembled with a program-synthesis approach (comparable to human performance). On BBH, TTT yields a 7.3% absolute improvement over few-shot prompting, achieving 57.8% accuracy. Gains are particularly large on tasks involving structural rules or distribution shifts (e.g., *Dyck languages*, *Ruin names*), where TTT yields 20–50 percentage points of improvement over standard in-context prompting.

Overall, our findings highlight that TTT drastically improves LM's few-shot learning ability on out-of-distribution tasks.

## 2. Preliminaries

### 2.1. In-context Learning

At a certain scale, many LMs exhibit the ability to adapt to new tasks without updating their parameters by simply con-

ditioning on input examples or instructions provided. Given a sequence of input-output pairs $(x_1, y_1), \ldots, (x_k, y_k)$ and a new input $x_{k+1}$, an LM can generate the corresponding output $\hat{y}_{k+1}$ by sampling from:

$$\hat{y}_{k+1} \sim \text{LM}(\cdot \mid x_1, y_1, \ldots, x_k, y_k, x_{k+1})$$

While the possibility of in-context learning (ICL) as implicit machine learning simulation is discussed in previous work (Akyürek et al., 2023), empirical evidence shows that in-context learning with language models does not always resemble standard machine learning algorithms (Zhao et al., 2024; Min et al., 2022b). Furthermore, ICL often struggles with novel tasks "out-of-the-box." For example, large language models exhibit poor performance on datasets like ARC (Opiełka et al., 2024; Bober-Irizar & Banerjee, 2024).

### 2.2. Test-Time Training

Test-time training (TTT) enables parametric models to adapt during inference through dynamic parameter updates in response to each test input. This approach remains relatively unexplored in the era of large language models. The general TTT process is as follows: starting with initial model parameters $\boldsymbol{\theta}_0$, for each test input (or batch of inputs) $d$, we generate a temporary training dataset $\mathcal{D}_{\text{TTT}}$. We then optimize these parameters to minimize a loss function

$$\arg\min_{\boldsymbol{\theta}} \sum_{d_{\text{TTT}} \in \mathcal{D}_{\text{TTT}}} \mathcal{L}(\text{LM}(d_{\text{TTT}}; \boldsymbol{\theta})),$$

resulting in temporarily updated parameters $\boldsymbol{\theta}_d$, which are subsequently used for prediction.[2]

In previous work (e.g., Sun et al., 2020), $\mathcal{D}_{\text{TTT}}$ is typically constructed by applying an unsupervised objective (e.g., masked autoencoding) to the input $x$ alone. In this paper, we extend TTT to the few-shot learning setting, treating it as a form of transductive learning by leveraging few-shot demonstration examples to improve predictions. Although TTT can also be applied to chain of thought (CoT; Wei et al., 2022), we focus on direct transduction, where demonstrations consist of input-output pairs $(x, y)$ without intermediate reasoning steps or explicit function descriptions.

The few-shot learning setting we consider provides richer context in the form of demonstration pairs $(x_1, y_1), \ldots, (x_K, y_K)$. One simple method for TTT is *Direct I/O* training, where we directly treat each input-output $(x_k, y_k)$ pair as training instances. Our key insight is that the few-shot examples can also be used to construct a more

---

[1]Code and data are available at https://github.com/ekina kyurek/marc (ARC) and https://github.com/adamzweiger /Fewshot-TTT (BBH).

[2]Note that this use of "test-time training" is related but distinct from the one used in recent line of work wherein an RNN's hidden state is treated as parameters and the update equation is interpreted as optimizing a recall-based regression objective (Ravi & Larochelle, 2017; Sun et al., 2024; Behrouz et al., 2025; Wang et al., 2025).

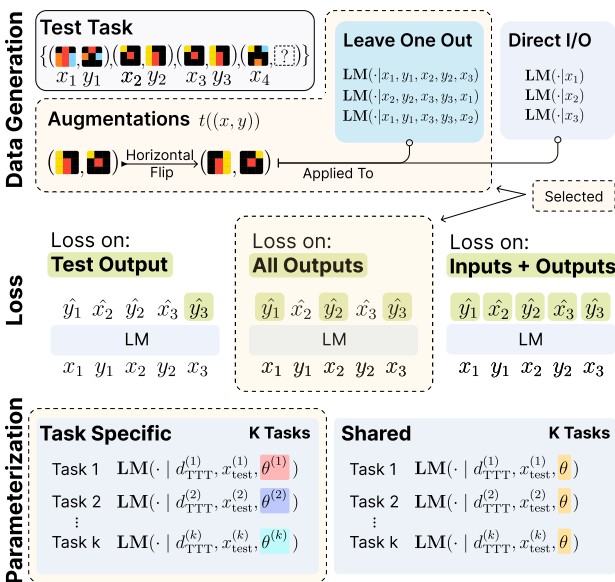

*Figure 2.* **TTT design decisions. Data generation:** A test task consists of input-output pairs $\{(x_i, y_i)\}$. The *Leave-One-Out* strategy removes one example at a time to form in-context learning tasks, while augmentations further expand the dataset. An alternative *Direct I/O* approach trains directly on the examples. **Loss:** The model is trained with loss computed on the *Test Output* (only the test-time prediction), *All Outputs* (including demonstration outputs), or *Inputs and Outputs* (all tokens). **Parametrization:** The *Task-Specific* approach trains a separate adapter per task while the *Shared* approach trains a single adapter across multiple tasks.

robust and expansive $\mathcal{D}_{\text{TTT}}$ of synthetic in-context learning tasks, allowing for effective model adaptation during test time. Additionally, when task-specific knowledge is available, this structure can be leveraged to further expand the dataset, as demonstrated in our experiments on ARC (Section 4). We also explore the general case where no task-specific information is used, as tested on BBH (Section 5).

Our experiments in this paper characterize each component of the TTT pipeline, investigating different design choices across the following stages: (1) constructing an input-specific training dataset $\mathcal{D}_{\text{TTT}}$ at test-time; (2) fine-tuning the LM by optimizing a loss function $\mathcal{L}$ over the dataset $\sum_{d \in \mathcal{D}_{\text{TTT}}} \mathcal{L}(\text{LM}(d; \boldsymbol{\theta}))$; and (3) sampling from the updated model with an augmented inference strategy based on self-consistency to obtain a final prediction.

## 3. TTT Design

This section discusses the key design choices and challenges of applying TTT to LLMs, including how to best leverage their in-context learning capabilities, how to structure data for effective processing, what optimization objective to use, and how to efficiently update model parameters. We detail these considerations in the construction of the TTT dataset and the optimization setup (Figure 2).

### 3.1. Data Generation

Given a task with $K$ training input-output pairs $\{(x_k, y_k)\}_{k=1}^K$, a *test-time training dataset* $\mathcal{D}_{\text{TTT}}$ can be created by either following an in-context learning setup or a direct input-output (direct I/O) setup (top row in Figure 2):

**Leave-one-out tasks** We begin with *leave-one-out* in-context learning tasks. For each pair $(x_j, y_j)$, we exclude it from the set of demonstrations and treat it as a "test" example within the newly formed synthetic task:

$$d_j^{\text{ICL}} = \Big(\{(x_k, y_k)\}_{k \neq j}, \, x_j, \, y_j\Big).$$

Here, $\{(x_k, y_k)\}_{k \neq j}$ serves as the "in-context demonstrations," and $(x_j, y_j)$ is the "synthetic test example." To increase the number of synthetic tasks, we additionally **permute the order** of the demonstrations in each $d_j^{\text{ICL}}$.

**Direct input-output (I/O) tasks** Rather than constructing in-context tasks, we treat each $(x_k, y_k)$ pair independently as a single training instance:

$$d_j^{\text{I/O}} = (x_j, y_j).$$

In this setup, the model is fine-tuned on these training pairs without in-context demonstrations. While this approach is more computationally efficient, our results (Sections 4 and 5) show that it underperforms methods that utilize in-context demonstrations.

**Data augmentation** For certain tasks with structured inputs (e.g., ARC), we can apply invertible transformations (e.g., flips, rotations, color permutations) to further augment the TTT dataset. Let $\mathcal{T}$ be a set of invertible transformations. For each $t \in \mathcal{T}$, we have $t^{-1}(t(x)) = x$, so we can apply $t$ to each training and test instance in $d_j$ to yield a transformed task $t(d_j)$. Since these transformations preserve the core relationships in the data (e.g., the input-output pattern is the same, just rotated), they effectively expand the training signal. If rule-based transformations are used, the final TTT dataset is: $\mathcal{D}_{\text{TTT}} = \bigcup_{t \in \mathcal{T}} \bigcup_j t(d_j)$.

### 3.2. Loss Function

We optimize the standard LM loss on $\mathcal{D}_{\text{TTT}}$. For the in-context leave-one-out setup, we experiment with 3 different ways to take the loss (middle row in Figure 2):

- **Test output (no demonstration loss)** The standard formulation where the loss is taken over $y_{\text{test}}$:

  $$\mathcal{L}_{\text{LM}}^{\text{label}} = \mathcal{L}_{\text{LM}}(y_{\text{test}} \mid x_1, y_1, \ldots, x_K, y_K, x_{\text{test}}; \boldsymbol{\theta})$$

- **All outputs**[3] In addition to the loss on the test output,

---

[3]For ARC, we start the indexing at $k = 2$ because the underlying transformation of an ARC task cannot be inferred without observing at least 1 demonstration.

the loss is also taken over the outputs of the in-context demonstrations, which encourages the model to correctly predict the demonstration outputs after seeing the previous demonstrations:

$$\mathcal{L}_{\text{LM}}^{\text{outputs}} = \mathcal{L}_{\text{LM}}^{\text{label}} + \sum_{k=1}^{K} \mathcal{L}_{\text{LM}}(y_k | x_1, y_1, ..., x_k; \boldsymbol{\theta})$$

- **Loss on inputs and outputs** The loss is taken over all tokens, encouraging the model to learn the structure of $x$ as well as $y$:

$$\mathcal{L}_{\text{LM}}^{\text{all}} = \mathcal{L}_{\text{LM}}^{\text{outputs}} + \sum_{k=1}^{K} \mathcal{L}_{\text{LM}}(x_k | x_1, y_1, \ldots, y_{k-1}; \boldsymbol{\theta})$$

This method, which requires learners to generate task *inputs* as well as outputs, is analogous to existing unsupervised TTT objectives (Sun et al., 2020).

We find in Sections 4.3 and 5.3 that the first method (taking the loss over both demonstration and test outputs) works best.

### 3.3. Parametrization

Once we have the test-time training dataset $\mathcal{D}_{\text{TTT}}$ (constructed via either the in-context or direct I/O approach), we perform a small number of gradient steps on task-specific **LoRA** adapters (Hu et al., 2022). This approach allows computationally efficient adaptation while maintaining the model's general capabilities. By default, we learn *task-specific* LoRA adapters for each ARC or BBH task at test-time. That is, we obtain $K$ different LoRA adapters, where $K$ is the number of test tasks. We also experiment with using a single *shared* LoRA adapter from the aggregated dataset of few-shot examples drawn from multiple tasks (bottom row in Figure 2)—a test-time version of meta-ICL (Min et al., 2022a). We find that the shared adapter degrades performance on ARC, whereas it improves performance on BBH. We discuss this in more detail in Section 5.3.

## 4. Abstraction and Reasoning Corpus

### 4.1. Background

The Abstraction and Reasoning Corpus (ARC) aims to evaluate the abstract reasoning capabilities of language models through their ability to solve visual puzzles. Each puzzle (henceforth referred to as a *task*) consists of input-output pairs of 2D grids (up to $30 \times 30$ in size) containing shapes or patterns in up to 10 different colors, as displayed in Figure 3. The output of each pair is obtained by applying an *intuitive* and *shared* transformation or rule $y = f(x)$. Each task has 2-7 demonstration examples and 1-3 test examples.

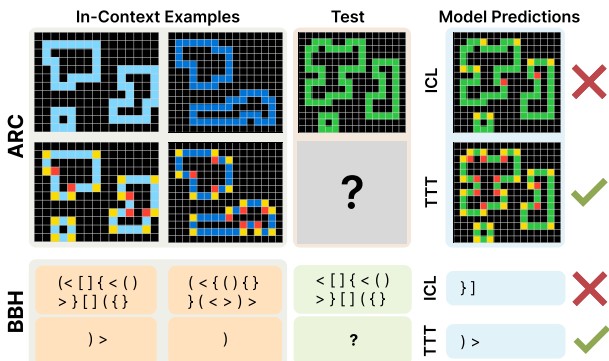

*Figure 3.* Example of ARC and BBH tasks that the model successfully solves only after applying TTT.

### 4.2. Experimental Details

**Model architecture & optimization** For our ablation experiments, we use the 1B-parameter Llama-3.2 model (Llama Team, 2024). For our final results in Section 4.6, we use the 8B Llama 3 model. We use Low-Rank Adaptation (LoRA; Hu et al., 2022) for parameter-efficient test-time training. More details are given in Appendix C.2.

**Fine-tuning before TTT** While TTT offers task-specific adaptation, the initial capabilities of the base model significantly influence its final performance (Section 4.4). We developed several approaches for generating synthetic training data to enhance the base model's abstract reasoning capabilities through fine-tuning, exploring both automated and semi-automated methods for task generation. This is complementary to TTT as the base model is fine-tuned on tasks distinct from those tested on, when TTT is applied. Details on our data generation strategies, as well as the effects of various data sources and model sizes on performance, are provided in Appendix B. The fine-tuned base model serves as the foundation for all subsequent experiments.

**Evaluation** The success criterion requires producing an exact match for all test outputs (no partial credit). Following the standard ARC scoring criteria, we use the pass@2 metric and produce 2 attempts for each test input. The original training and validation sets consist of 400 tasks each. However, for efficient evaluation purposes, we randomly pick 80 balanced ARC tasks from the ARC validation set, including 20 easy, 20 medium, 20 hard, 20 expert tasks according to the classification in (LeGris et al., 2024) (see Appendix A.2 for this task list). Except for our final results, we use this subset of ARC tasks throughout our experiments. We limit $\mathcal{D}_{\text{TTT}}$ to have a maximum of 250 examples per task for efficiency reasons. Appendix C.2 provides additional details on the hyperparameters.

**Inference** One of the most common techniques to scale inference-time compute is to use temperature sampling to

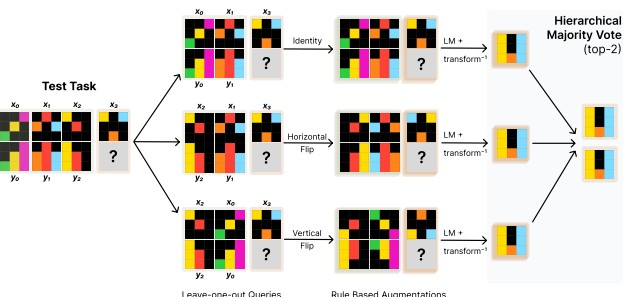

*Figure 4.* **Augmented inference and hierarchical voting.** We use leave-one-out tasks and invertible geometric transformations to obtain multiple equivalent versions of the task for augmented inference. Predictions from these versions are aggregated with a hierarchical voting strategy: first, voting is performed within each transformation, and then the top candidates from each transformation undergo global voting to yield the top two predictions.

obtain multiple responses and select the best according to a ranker, called self-consistency (Wang et al., 2023). However, this is not viable in ARC (where the output grid is directly predicted) as there is no way to directly enforce diversity *across* samples while ensuring coherence *within* samples. As an alternative self-consistency approach, we try an *augmented inference* strategy that combines greedy decoding with multiple versions of the input. Specifically, we generate multiple prediction candidates by using geometric transformations. We then employ a hierarchical voting strategy to determine the final prediction from the set of generated candidates. This approach involves two stages of voting to progressively narrow down the best candidates: (1) **Intra-transformation voting:** Group predictions by their corresponding transformation $t$. Within each group, select the top-3 most frequent predictions. (2) **Global voting:** Take the selected transformation-specific candidates from the previous step and select the top-2 most frequent predictions *across* all transformations. The augmented inference pipeline is summarized in Figure 4 and full details of the pipeline are in Appendix E.

### 4.3. Impact of TTT Design

In this section, we compare the final implementation of our method with different design choices for TTT. **FT** serves as the baseline, using only the fine-tuned model with demonstrations in-context. **No Transformations** omits the augmentation step. **Direct I/O Data** replaces in-context tasks with the direct input-output task formulation (Section 3.1). **Shared TTT** uses a single LoRA adapter across all tasks instead of learning one per task. **No Demonstration Loss** removes the loss on demonstration outputs (Section 3.1).

Results are presented in Figure 5. Our TTT method is effective, improving fine-tuned model accuracy approximately **6×** (**5%** → **29%**). In-context formatting is especially im-

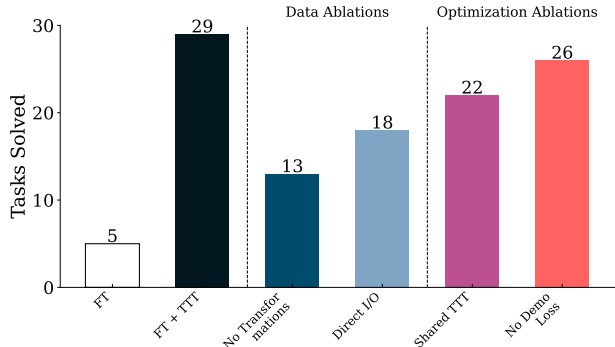

*Figure 5.* **Accuracy of different data and optimization ablations in TTT on ARC.** Our data ablations reveal that the ICL data format is crucial for effective TTT, and that applying transformations to augment the TTT dataset notably enhances performance. For optimization, learning task-specific adapters significantly outperforms using a single adapter and taking a loss on the in-context demonstrations provides a minor performance boost.

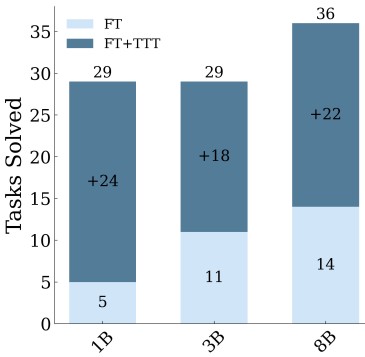

*Figure 6.* **Performance results across model sizes.** Fine-tuned model performance improves with increasing size. However, the scaling behavior after TTT is less clear. For instance, the final performance of the 1B and 3B models is identical after TTT.

portant; using the direct input-output data to construct $\mathcal{D}_{\text{TTT}}$ causes an 11-task drop (38%). Removing transformations causes a 16 task drop (55%). Regarding optimization, per-task LoRA adapters outperform a single shared adapter by 7 tasks (24%). Including losses on the demonstration outputs yields a modest but consistent gain (26% → 29%).

### 4.4. Impact of Model Size

We perform full fine-tuning of 1B and 3B Llama 3.2 (instruction-tuned) and 8B Llama 3 (instruction-tuned) using synthetically generated data, as detailed in Appendix B, and then use our default TTT implementation. We show results using different model sizes in Figure 6. Increasing the model size consistently improves FT performance, with the 8B model achieving the highest accuracy of 36%. At all model sizes, TTT leads to significant improvements in performance. We also observe that for smaller model sizes, TTT effectively closes the performance gap, with the 1B and 3B models achieving similar accuracy after TTT.

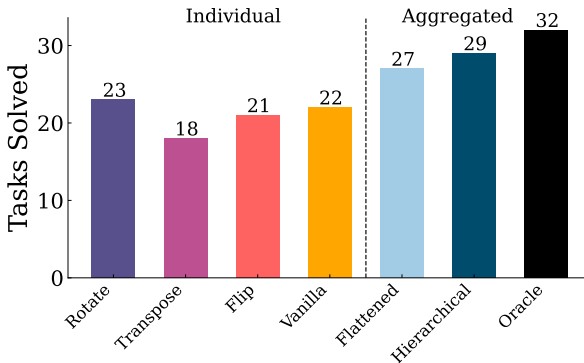

*Figure 7.* **Accuracy of different transformations and voting schemes.** While individual transformations generally perform at a modest level and are comparable to one another, aggregating across them through voting yields substantial improvements. Notably, a hierarchical voting strategy with two voting stages surpasses a flat voting approach. Our hierarchical method approaches oracle-level performance, demonstrating its effectiveness in accurately selecting the correct answer when present.

### 4.5. Impact of Augmented Inference

To analyze the impact of augmented inference and voting, we run several ablations: (1) **Vanilla**, which generates two predictions without transformations or advanced voting; (2) **Transformed Inference**, applying a single transformation (Rotate, Transpose, or Flip) to measure its isolated effect; (3) **Hierarchical Voting**, our full pipeline combining augmented inference and structured voting; (4) **Flattened Voting**, which selects the top-2 predictions from a single voting round over all generated outputs; and (5) **Oracle**, an upper bound that selects the correct answer if present.

As shown, individual transformations are modestly effective on their own (with *Transpose* performing worst), but their aggregation improves results markedly. Hierarchical voting further outperforms a flattened voting approach and closely approaches the oracle's accuracy, suggesting that our two-stage aggregation effectively identifies the correct solution when it is present.

### 4.6. Comparison to Other Systems

Following our experiments on 80 tasks, we present comprehensive results on the **full** ARC public evaluation set, comparing our system against existing approaches. Our analysis focuses on three key aspects: the impact of our TTT methodology, the benefits of combining our approach with existing methods, and the differences between fully neural and program synthesis methods.

**TTT** We apply TTT and augmented inference procedure to our base fine-tuned model (fine-tuned 8B model). TTT significantly improves accuracy from 18.3% to 47.1%.

**Integration with existing methods** A concurrent work by Li et al. (2025) introduced BARC, achieving 54.4% accu-

| PS | Fine-tuned LM | TTT Method | Score |
|---|---|---|---|
| X | Ours | X | 18.3% |
| X | Ours | Ours | 47.1% |
| X | BARC | Ours | 53.0% |
| BARC | Ours | Ours | 58.5% |
| BARC | BARC | Ours | **62.8%** |
| | | Avg. Human | 60.2% |
| | | Best Human | **97.8%** |
| | | BARC (ensemble) | 54.4% |
| | | BARC (no synthesizer) | 39.3% |
| | | Claude 3.5 Sonnet | 21.0% |
| | | GPT-4o | 9.0% |
| | | OpenAI o1-preview | 21.0% |
| | | DeepSeek r1 | 20.5% |
| | | OpenAI o3-preview | **82.8%** |

*Table 1.* **Pass@2 Scores of different systems on the ARC validation set.** Our TTT pipeline improves base models consistently. We achieve 47.1% accuracy when applied to our fine-tuned model and 53.0% when applied to the BARC model (Li et al., 2025). We ensemble our method with program synthesis (PS) based models, where we achieve score of 61.9%, comparable to the average human performance of 60.2%.

racy by combining neural and program synthesis approaches. While their fully neural approach shares similarities with our system, our TTT and inference pipeline has several additional components (per-task LoRA, more augmentations, hierarchical voting) that boost performance. To validate our improvements, we applied our TTT pipeline to BARC's fully neural model, achieving 53.0% accuracy—a 35% improvement over their original TTT method.

Building on these results, we explored combinations of our approach with BARC. Combining our TTT pipeline and neural model with BARC's synthesizer raised accuracy to **58.5%**. Combining our TTT pipeline with BARC's neural model and synthesizer raised accuracy to **61.9%**. This configuration matches average human performance of 60.2% (LeGris et al., 2024) on the benchmark.

**Comparing program generation and end-to-end modeling** Li et al. (2025) found that program synthesis and fully neural predictors for ARC are highly complementary. Their end-to-end neural model can only solve 42.2% of the tasks solved by the program synthesis model. However, we find that when equipped with our TTT pipeline, BARC's fine-tuned fully neural model solves 73.5% of the tasks that are solved by the program synthesis model. This suggests that our TTT pipeline significantly improves the neural model's ability to learn systematic reasoning patterns similar to those captured by program synthesis models.

**Semi-private evaluation** ARC-AGI challenge provides a hidden "semi-private dataset" and performs external tests for submissions. We submitted our ensemble solution to the official ARC-AGI semi-private evaluation and observed 47.5% accuracy. This decline may be attributed to more

significant distribution shifts in the semi-private evaluation dataset. A more detailed analysis of these performance differences can be conducted in the future once the semi-private set is publicly released.

# 5. BIG-Bench Hard

## 5.1. Background

BIG-Bench Hard (BBH; Srivastava et al., 2023; Suzgun et al., 2023) is a benchmark comprising 27 challenging tasks across 23 task types, designed to evaluate large language models on reasoning, compositionality, and generalization. Unlike ARC, BBH features a broader natural language structure and lacks a shared input format, making it unsuitable for invertible transformations. However, this broader scope offers a valuable testbed for evaluating TTT's effectiveness in a more generalized setting. Despite the absence of invertible transformations—previously used in ARC to expand the TTT dataset and enhance inference—TTT still significantly improves performance on BBH.

## 5.2. Experimental Details

**Model architecture & optimization** We use Llama 3.1 (8B; Llama Team, 2024). For each task $d$, we train a separate set of LoRA parameters at test-time, with a LoRA rank of 64 over 40 random shuffles of the demonstration pairs to produce leave-one-out in-context tasks. More hyperparameter details are given in Appendix F.1.

On BIG-Bench Hard, our base language model is able to achieve non-trivial scores out-of-the-box. Consequently, we do *not* perform any initial fine-tuning on synthetic tasks outside of BBH like we do for ARC. Furthermore, since models achieve nonzero performance in a zero-shot setting, we provide the zero-shot results and analyze how TTT and ICL improve upon them.

**Evaluation** For the 27 tasks in BBH, we consider the 10-shot setting, where we select 10 random pairs from each task's dataset to be demonstration pairs and evaluate on the remaining data. Each of the 27 tasks is analogous to a single ARC task, consisting of 10 labeled examples as demonstration pairs given at test-time. We report average results over five random seeds, where each seed specifies which 10 examples form the demonstration subset. For more control over the evaluation process with test-time training, we write our own evaluation function, which is available in our codebase (for more details, see Appendix F.1). The number of evaluation examples for each task is then 240 for all tasks except three: *Causal Judgment*, *Penguins in a Table*, and *Snarks*, which have 177, 136, and 168 evaluation examples respectively. Note that the large number of evaluation samples for each task compared to ARC means we can do a task-specific analysis to analyze which types of tasks

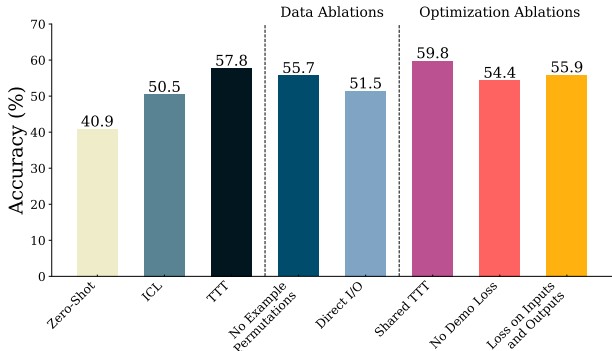

*Figure 8.* **Overall BIG-Bench Hard Results.** TTT outperforms standard in-context learning by 7.3 absolute percentage points, from 50.5% to 57.8%. Our performance improvement over direct input-output data shows that using in-context leave-one-out tasks is crucial. Not taking demonstration loss or taking loss on inputs results in a performance decrease. Unlike with ARC, using a shared adapter across all tasks improves performance.

benefit the most from TTT (Section 5.4). Unlike with ARC, we do not have a collection of invertible transformations to run augmented inference. Instead, we use greedy decoding. Further hyperparameter details and evaluation details are given in Appendix F.2.

## 5.3. Impact of TTT Design

In this section, we evaluate our method and its ablations, primarily comparing the zero-shot baseline, **ICL**, and **TTT**. **No Example Permutation** updates the model on a single in-context prompt instead of multiple shuffled versions. **Direct I/O** treats each input-output pair as separate training instances. **Shared TTT** uses a single adapter across tasks instead of task-specific adapters. **No Demonstration Loss** removes the loss applied to demonstration outputs. **Loss on Inputs and Outputs** extends the loss calculation to both inputs and outputs. These ablations are as detailed in Section 3. As these results are averages over 5 runs, the standard errors of the mean for each method are given in Appendix F.1, averaging 0.4%.

The results in Figure 8 show that TTT achieves an overall accuracy of 57.8%, outperforming standard ICL (50.5%) and Direct I/O learning (51.5%). This demonstrates that TTT's capabilities extend beyond ARC to more diverse and complex reasoning tasks, proving its effectiveness in a broader range of natural language problem-solving scenarios.

We observe that TTT without example permutations—performing multiple gradient steps on a single in-context prompt before inference—reduces accuracy to a still-impressive 55.7%. Computing the loss only on the test output lowers accuracy to 54.4%, while applying it to both inputs and outputs achieves 55.9%.

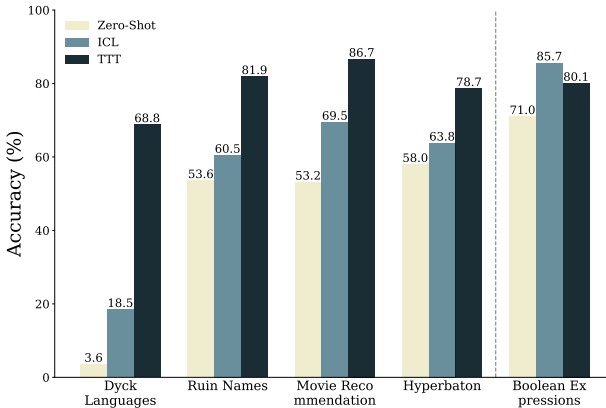

*Figure 9.* **BIG-Bench Hard results for tasks with the largest TTT-ICL score differences.** The four tasks on the left show the most significant improvements with TTT over ICL, while the task on the right has the lowest TTT score relative to ICL. Full task-specific results are given in Appendix F.2.

**Shared adapter**   Unlike on ARC, using a shared adapter *improves* performance on BBH, indicating that tasks in BBH do not confound each other during training. On the ARC dataset, each puzzle has the same input format, so distinguishing among multiple tasks is difficult, and we may have conflicting gradients with a single adapter. In BBH, however, distinguishing tasks is trivial (the instructions differ in plain text), and many tasks are mutually helpful. For instance, updating on *Logical Deduction Five Objects* also aids *Logical Deduction Three Objects*, without hurting *Word Sorting*. Although this is no longer test-time training on distinct tasks presented individually at test time, it can be interpreted as TTT on the entire dataset presented collectively at test time.

### 5.4. Task-Specific Analysis

Our task-specific results show that performance improvements from TTT are highly *task-dependent*. Among the 27 tasks in BBH, TTT results in a performance decline of at least 2% compared to ICL in only 2 tasks. In contrast, 12 tasks show an improvement of at least 2%, with 9 of these showing improvements of at least 5%. The four tasks with the most significant performance boost from TTT over ICL or zero-shot and the task with the most significant performance decrease are shown in Figure 9. These tasks in order of TTT's improvement over ICL are *Dyck Languages* (parentheses matching), *Ruin Names* (humorous name modifications), *Movie Recommendation* (choosing similar films), *Hyperbaton* (adjective ordering), and *Boolean Expression* (evaluating a boolean expression). Detailed results for every task are given in Appendix F.2.

We hypothesize that improvements from TTT may be driven by tasks involving distribution shifts and structured patterns. For example, tasks like *Dyck Languages* and *Hyperbaton* follow clear grammatical or programmatic rules, which could

align well with TTT's ability to adapt to latent structural regularities during test-time.

Conversely, tasks requiring explicit step-by-step computation show limited gains with TTT. For instance, *Boolean Expressions* declined from 85.7% to 80.4% under TTT. This task's algorithmic nature—dependent on sequential reasoning rather than pattern-based transduction—and its likely pre-training exposure suggest TTT's updates may not resolve its specific demands. While these particular observations align with our hypothesis, the reason certain tasks benefit more from TTT remains an open question.

## 6. Related Work

**Test-time training**   The idea of updating model parameters at test-time using instance-specific data traces back to early work on local learning (Bottou & Vapnik, 1992). More recently, Sun et al. (2020) propose a simple test-time self-supervision scheme to adapt an image classifier when facing distribution shifts. In language modeling, Hardt & Sun (2024) fine-tune on retrieved neighbors at test-time for notable gains, while Hübotter et al. (2025) optimize retrieval via active data selection.

**ARC challenge**   Abstraction and Reasoning Corpus (ARC; Chollet, 2019; Chollet et al., 2025) is a collection of extremely challenging few-shot visual reasoning problems. Most approaches to ARC fall into two main categories: *program synthesis* and *fully neural*. Program synthesis approaches (Butt et al., 2024; Wang et al., 2024; Li et al., 2025; Greenblatt, 2024) first try to find the transformation function $f$, and then apply it to the test example. Fully neural approaches (Veldkamp et al., 2023; Bober-Irizar & Banerjee, 2024) try to directly predict the output $y^{\text{test}}$, only implicitly modeling $f$. In this work, we use a fully neural approach, using an LM to predict the test outputs. Recent work has explored hybrid methods, leveraging inference scaling and deep learning-guided program synthesis (Greenblatt, 2024; Li et al., 2025). Similarly, we find that integrating our neural model with program synthesis improves performance.

## 7. Conclusion

We conduct an investigation of test-time training and demonstrate that it can significantly improve LM performance on abstract reasoning and few-shot learning tasks, namely the Abstraction and Reasoning Corpus (ARC) and BIG-Bench Hard (BBH). Our key contributions include a robust TTT framework with leave-one-out in-context task construction, the optimization setup, and the inference strategy after TTT. Our results reveal the potential of TTT to tackle novel reasoning tasks, suggesting significant promise for test-time methods in advancing the next generation of LMs.

## Limitations

**Optimization bias**  In development of ARC, we used a set of 80 tasks for validation/ablation experiments. Standard hyper-parameters (learning rate, epochs) were optimized using this set, which might have introduced some bias.

**Data leakage**  While the base Llama-3 performs poorly on the public validation set of ARC, the public availability of the dataset introduces the possibility that these models may have seen these examples during pre-training. Similarly, while the base model achieves reasonable performance on BBH, its public availability raises similar concerns.

## Acknowledgments

We sincerely thank the BARC team (Li et al., 2025) for their support and collaboration in ensembling our method with theirs, resulting in an official joint submission to the ARC public set. We thank Aniruddha Nrusimha for helpful discussions on parameter efficient training. This work was supported by MIT–IBM Watson AI Lab, the MIT Quest for Intelligence, and by the National Science Foundation under grants IIS-2212310, IIS-2238240, and CCF-2217064. JA is additionally supported by a Sloan Research Fellowship. This work also benefited from many conversations during the Simons Institute Program on Language Models and Transformers.

## Impact Statement

This paper presents work whose goal is to advance the field of machine learning. There are many potential societal consequences of our work, none which we feel must be specifically highlighted here.

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

# A. ARC Dataset

We present the tasks in the development set, the data format and evaluation details for the ARC dataset (available at this https link.).

## A.1. Data Format

We use numpy's array printing format for all experiments as shown in Figure 10.

## A.2. List of 80 Tasks Used For Development

We use the following (Table 2) tasks validation tasks for our development.

Table 2. Selected development tasks and their hardness level based on (LeGris et al., 2024).

| ID | Level | ID | Level | ID | Level | ID | Level |
|---|---|---|---|---|---|---|---|
| 0a1d4ef5 | easy | 762cd429 | medium | e5c44e8f | hard | e99362f0 | expert |
| 692cd3b6 | easy | e7639916 | medium | 604001fa | hard | 1acc24af | expert |
| 1da012fc | easy | e1d2900e | medium | 4364c1c4 | hard | f9a67cb5 | expert |
| 66e6c45b | easy | aee291af | medium | 506d28a5 | hard | ad7e01d0 | expert |
| 3194b014 | easy | e95e3d8e | medium | 2037f2c7 | hard | ea9794b1 | expert |
| 963f59bc | easy | e0fb7511 | medium | d5c634a2 | hard | 58e15b12 | expert |
| d37a1ef5 | easy | ae58858e | medium | ac605cbb | hard | 891232d6 | expert |
| 358ba94e | easy | 93c31fbe | medium | 27f8ce4f | hard | 5833af48 | expert |
| f3cdc58f | easy | 27a77e38 | medium | 66f2d22f | hard | 4ff4c9da | expert |
| 55059096 | easy | 9bebae7a | medium | 3ed85e70 | hard | 5b692c0f | expert |
| c7d4e6ad | easy | 9ddd00f0 | medium | 8b28cd80 | hard | e2092e0c | expert |
| 4b6b68e5 | easy | fe9372f3 | medium | d19f7514 | hard | 47996f11 | expert |
| 00576224 | easy | 69889d6e | medium | dc2aa30b | hard | 34b99a2b | expert |
| a04b2602 | easy | 15663ba9 | medium | f5c89df1 | hard | 1c56ad9f | expert |
| e9c9d9a1 | easy | 17b80ad2 | medium | 50f325b5 | hard | e6de6e8f | expert |
| ef26cbf6 | easy | 16b78196 | medium | 08573cc6 | hard | fea12743 | expert |
| 7ee1c6ea | easy | 5b6cbef5 | medium | 3d31c5b3 | hard | 31d5ba1a | expert |
| e9ac8c9e | easy | 40f6cd08 | medium | 94133066 | hard | 79fb03f4 | expert |
| 1a2e2828 | easy | 505fff84 | medium | 136b0064 | hard | 8719f442 | expert |
| 770cc55f | easy | d017b73f | medium | 90347967 | hard | a8610ef7 | expert |

## A.3. Evaluation

We follow the competition rules that if any of the two pass@2 predictions of the system is correct, we consider that test correct. In the reported task-level accuracies, we did not give partial points if all tests are not solved, except the final table Section 4.6.

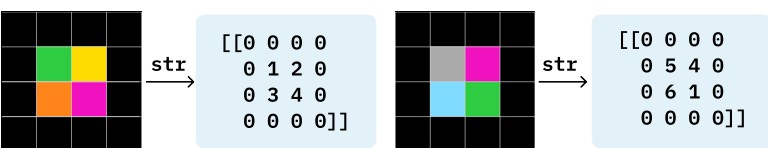

Figure 10. **Data Format:** We convert grids to strings by representing them as numpy arrays of digits from 0 to 10 where each digit corresponds to a different color.

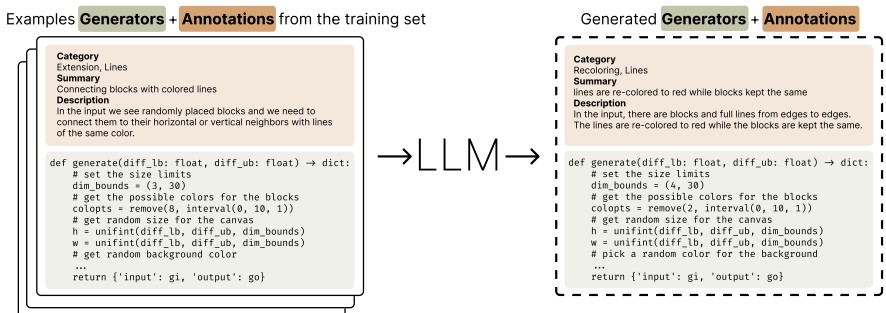

*Figure 11.* **LLM based synthetic tasks generation:** Given some seed task descriptions and task generator functions in Python, we generate more generator functions to produce novel tasks. We use three different approaches: (1) few-shot prompting with only generators, (2) few-shot prompting with generators and task descriptions, (3) two-stage approach: first generate free form descriptions, then condition on them to generate more generators (shown in Figure 12).

# B. Fine-Tuning Before TTT

While test-time training facilitates task-specific adaptation, the base model's capabilities impacts the final performance. We developed several approaches for generating synthetic training data to enhance the base model's abstract reasoning capabilities through fine-tuning, exploring both automated and semi-automated methods for task generation. In this section, we detail our fine-tuning data generation strategies and analyze the impact of different data sources and model sizes on final performance.

### B.1. Preparing Fine-tuning Data

(Hodel, 2024) provides domain-specific language (DSL), REARC, as well as the transformation $f_i$ that solves the task-$i$, and the data generation function $g_i$ that are implemented in this DSL for each training task in the $\mathcal{D}_{\text{ARC}}^{\text{train}}$ dataset. These functions enable sampling of new input-output pairs that maintains the same underlying transformation principle:

$$d = (x, y) \sim \text{eval}(g_i) \tag{1}$$

where $d$ represents a newly generated input-output pair that can be solved using the same transformation function $f_i$ as the original task-$i$[4].

**(a) Using Existing Generators**   The generator functions $g$ in REARC already provide an effective data augmentation tool by producing different instantiations of same tasks. We generate extra samples from these training tasks by running the code many times and randomly splitting these new examples ($d \sim \text{eval}(g_i)$) to a set of train and test examples. These augmented examples are already provided with their DSL release.

**(b) Few-shot Prompting an LLM**   Additionally, we used several approaches to generate *novel* tasks using an LM (in our case, an ensemble of GPT4 and GPT4-o).

The simplest approach generates new task generators using few-shot examples:

$$g' \sim \text{LM}(g_1, g_2, \ldots, g_m) \tag{2}$$

where $g'$ is a new generator function and $g_1, \ldots, g_m$ are existing generator functions (shown in Figure 11). We sample different $m$ examples by uniformly from existing training set. We repeat this process multiple times to get a good amount of tasks.

We augment the generator functions with task descriptions and jointly generate both descriptions and generators:

$$(s', g') \sim \text{LM}(s_1, g_1, s_2, g_2, \ldots s_m, g_m) \tag{3}$$

where $s_i$ represents the description of task $i$.

---

[4]We can verify the generated examples by asserting $f_i(x) = y$.

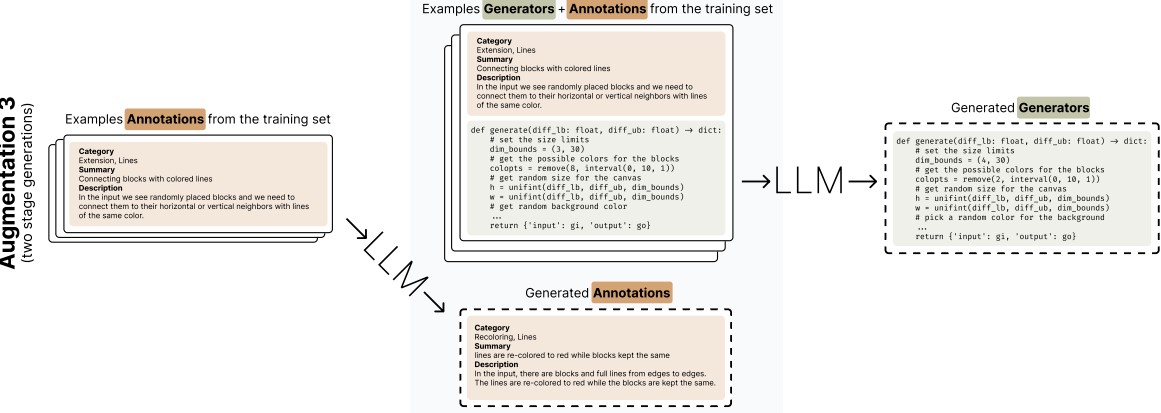

*Figure 12.* **Two-stage generation using an LLM**: First, we prompt the LLM to generate a task description using few-shot prompting. Then, we generate the new generator based on existing task pairs and the newly created description.

To get the task descriptions, we manually created seed descriptions for 10 training tasks. These seed descriptions were then used to generate descriptions for the training and validation tasks through few-shot prompting. To increase diversity of tasks, we use task descriptions with hierarchical fields (category, summary, and description). The process of getting these descriptions is provided in Appendix D.1.

Instead of jointly generating task descriptions and function generations, we additionally deployed a two-stage approach (Figure 12 ) described as following:

$$s' \sim \text{LM}(s_1, s_2, \ldots s_m) \tag{4}$$

$$g' \sim \text{LM}(s_1, g_1, s_2, g_2, \ldots, s_m, g_m, s') \tag{5}$$

This approach first generates a task description $s'$ and then conditions the generator creation on both existing task pairs and the new description. In total we collected 6426 generators with these LLM based approaches. We provide qualitative samples from these LM generated tasks in Figure 16.

**(c) Geometric Transformations** Finally, our synthetic tasks are enhanced through various geometric transformations, such as basic transformations (rotations, reflections, random shift and size scaling), pattern operations (random patching, tiling, and repetition), color permutations, and composite transformations involving sequential application of multiple basic transformations. These transformations are applied in three ways:

- Input grids only: $(x, y) \rightarrow (t(x), y)$
- Output grids only: $(x, y) \rightarrow (x, t(y))$
- Both input and output: $(x, y) \rightarrow (t(x), t(y))$

We use all the transformations given in Appendix C.1, and some additional transformations given in Table 3. In the fine-tuning case, different from TTT, we apply augmentations to only inputs, only outputs or both. These transformations are applied randomly to variants of tasks with 30% of the time.

*Table 3.* We provide the additional augmentations use in our data generation for fine-tuning with their function signature and description.

| Augmentation Name | Description |
| --- | --- |
| Repeat(direction, n) | Rotates a grid in horizontal or vertical direction by $n$ times. |
| DropoutOutput | Randomly deletes some patches of the output grids. |
| DropoutInput | Randomly deletes some patches of the input grids |

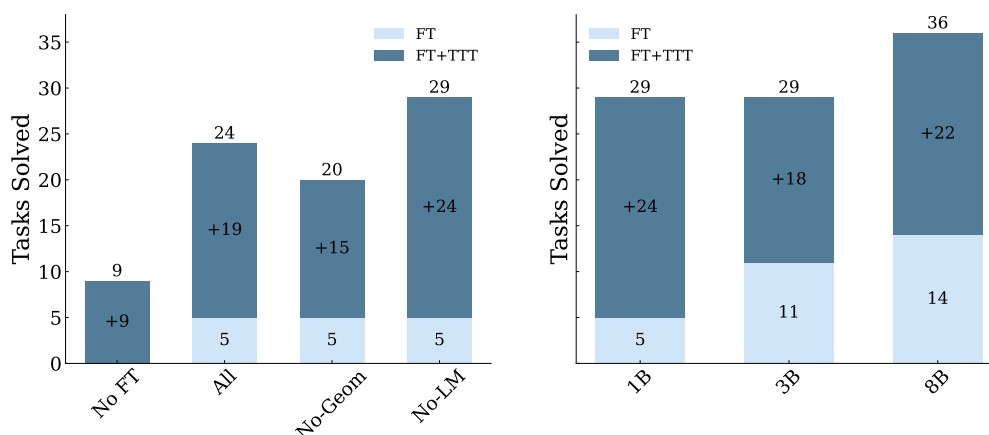

*Figure 13.* **Left: Accuracy when fine-tuning with different data sources.** While all fine-tuned models perform similarly, their performance after TTT shows considerable variance. As expected, removing geometric transformations from the fine-tuning data reduces performance compared to the model trained on the full dataset. Surprisingly, excluding LM-generated data from fine-tuning actually outperforms the model trained on all data. **Right: Performance results across different model sizes.** As expected, performance of the base fine-tuned model improves with increasing model size, aligning with current scaling law trends. However, the scaling behavior after TTT is less clear. For instance, the final performance of the 1B and 3B models is identical after TTT. Full discussion in Section B.3.

### B.2. ARC Initial Fine-tuning Hyperparameters

We perform full fine-tuning on LLama-3 family models by using the `torchtune` library. We train each model up to 16000 steps. We use 2xNVIDIA A100 GPU for 1B models, 4xNVIDIA A100 GPU for 3B and 8B models. We present hyperparameters in Table 4.

*Table 4.* ARC Initial Fine-tuning Hyperparameters

| Hyperparameter | Search Space |
| --- | --- |
| learning rate | 2.5e-5 |
| epochs | 2 |
| batch size | 32 |
| optimizer | AdamW (Loshchilov & Hutter, 2018) |
| scheduler | Cosine LR Schedule with 2k warmup |

### B.3. Results

We perform full fine-tuning 1B, 3B Llama 3.2 instruction-tuned, and 8B Llama 3 instruction-tuned using augmented data. The format and training objective is same as the ones described for TTT in 3. Hyperparameter details are given in Appendix C.2. We do the following ablations for augmented data:

1. **No FT:** The original Llama 3 instruction-tuned model without any fine-tuning.

2. **All:** We use all methods described in Section B.1, including REARC, rule-based augmentation, and LM generation.

3. **No-Geom:** We remove geometric transformations from all tasks.

4. **No-LM:** We only use REARC and rule-based augmentation, excluding tasks generated by the LM.

We show results using different model sizes in Figure 13. Increasing the model size consistently improves FT performance, with the 8B model achieving the highest accuracy of 36%. We also observe that TTT effectively closes the performance gap for smaller models, with the 1B and 3B models achieving similar accuracy after TTT.

# C. TTT Transformations for ARC

We present the transformations used in TTT and the training details.

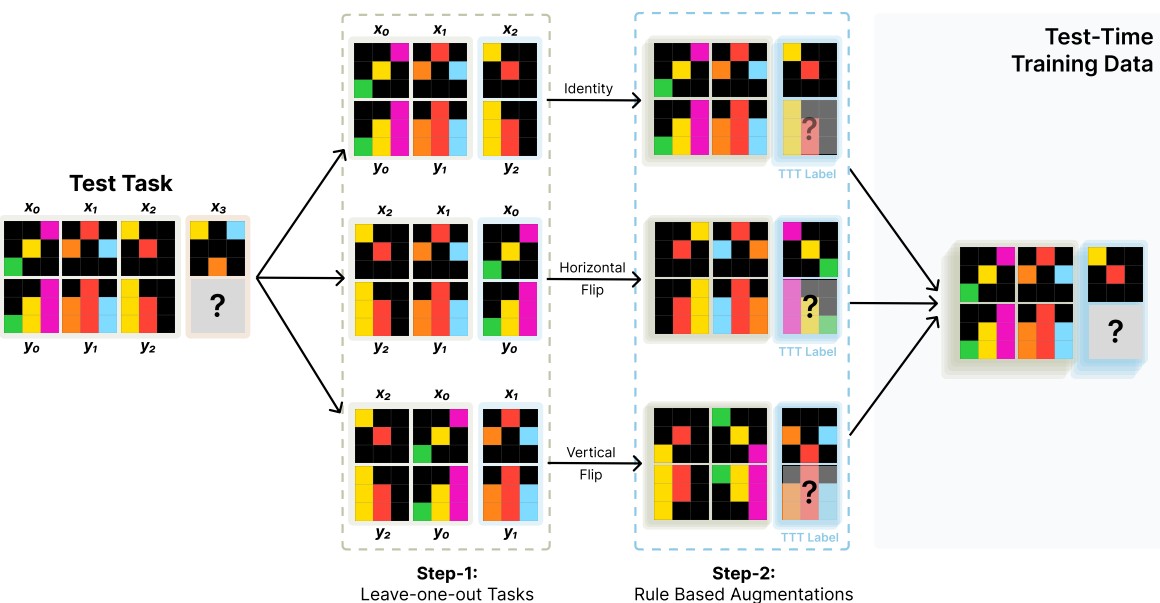

*Figure 14.* **TTT dataset generation for a test task (Section 3.1):** We start by creating leave-one-out tasks from the given training examples of the task. These tasks are then augmented through rule-based transformations to obtain the full TTT dataset. Finally, we train task-specific LoRA adapters on top of the base FT model.

## C.1. Transformations

We provide the augmentations used in TTT in Appendix C.1, please refer to our code base for their implementations. After applying these augmentations, we additionally shuffle colors and shuffle training examples. Note that these transformations are applied to all input and output grids. The procedure for generating the dataset for TTT is shown in Figure 14.

## C.2. Training Setup & Hyperparameters

We use the `torchtune`(torchtune Maintainers & Contributors, 2024) library to train LoRA adapters on Llama-3 family of models. We apply LoRA training to query and value projection weights of the self-attention layer, to the MLP weights and to the output projection layer (was only available for Llama-3 8B in `torchtune`). We present hyperparameters of this training in Table 6. We also found that using quantized LoRA adapters (Dettmers et al., 2023) instead of standard (full-precision) LoRA leads to only a small drop in performance ($29 \rightarrow 26$ tasks solved with the 1B-parameter model), making it a viable option in memory-constrained settings.

We resort to the `vLLM` (Kwon et al., 2023) library for prediction as it provides fast kernels and batched inference for our models and LoRA inference. We just use greed decoding as we did not see improvements with temperature sampling in our early experiments. We use 90, 180 degree rotations, horizontal, vertical, and diagonal (transpose) flips as our invertible transformations.

With that, the whole TTT and inference process takes approximately 12 hours for 100 randomly sampled validation tasks when using an NVIDIA A100 GPU.

*Table 5.* We provide the augmentations used in our TTT procedure with their function signature and description.

| Augmentation Name | Description |
|---|---|
| `Rotate(90)` | Rotates a grid 90 degrees. |
| `Rotate(270)` | Rotates a grid -90 degrees. |
| `Rotate(180)` | Rotates a grid 180 degrees. |
| `Flip(0)` | Flips a grid horizontally |
| `Flip(1)` | Flips a grid vertically |
| `Reflect(0, reverse=True)` | Flips a grid horizontally and prepends to the left of the original grid |
| `Reflect(1, reverse=True)` | Flips a grid vertically and prepends to above the original grid |
| `Reflect(0, reverse=False)` | Flips a grid horizontally and appends to the right of the original grid |
| `Reflect(1, reverse=False)` | Flips a grid vertically and appends to below the original grid |
| `RandomTranslateXY()` | Shifts a grid randomly in the horizontal and vertical directions. The maximum shift size is 4 |
| `Transpose()` | Reflects a grid on diagonal |
| `IncreaseResolution(2)` | Upscales the grid by interleaving elements in both horizontal and vertical directions |
| `IncreaseHeight(2)` | Upscales the grid by interleaving elements in vertical direction |
| `IncreaseWidth(2)` | Upscales the grid by interleaving elements in horizontal direction |
| `Chain([Rotate(90),IncreaseResolution(2)])` | Sequential application of `Rotate(90)` and `IncreaseResolution(2)` |
| `Chain([Rotate(270),IncreaseResolution(2)])` | Sequential application of `Rotate(270)` and `IncreaseResolution(2)` |
| `Chain([Rotate(180),IncreaseResolution(2)])` | Sequential application of `Rotate(180)` and `IncreaseResolution(2)` |
| `Chain([Flip(0),IncreaseResolution(2)])` | Sequential application of `Rotate(180)` and `IncreaseResolution(2)` |
| `Chain([Flip(1),IncreaseResolution(2)])` | Sequential application of `Rotate(180)` and `IncreaseResolution(2)` |
| `Chain([Transpose(),IncreaseResolution(2)])` | Sequential application of `Rotate(180)` and `IncreaseResolution(2)` |

*Table 6.* ARC TTT Hyperparameters. We find learning rate of 5e-5 the best for 1B and 3B models, and 1e-4 the best for 8B models.

| Hyperparameter | Search Space |
|---|---|
| $r$ LoRA rank | 128 |
| $\alpha$ LoRA alpha | 16 |
| learning rate | [**5e-5**, **1e-4**] |
| epochs | 2 |
| batch size | [1, **2**] |
| optimizer | AdamW (Loshchilov & Hutter, 2018) |

# D. LM Data Generation

We described three approaches in Appendix B to use LM, we generated 6426 task generators by few-shot prompting GPT-4 and GPT-4o models (OpenAI, 2024; Hurst et al., 2024).

## D.1. Getting Descriptions for Tasks

This procedure is shown in Figure 15. We initially described 10 training tasks with the hierarchical-style shown in Figure 11. Then, for other training tasks tasks, we obtained less quality crowd-worker annotations from LARC (Acquaviva et al., 2022) project. By using our high-quality seed annotations and their LARC version, we 10-shot prompt and LM to produce high quality annotations for the other training tasks.

---

You are an intelligent agent that can induce task descriptions from examples. For Category, please \*do not\* use generic terms like Transformation, Pattern Recognition.
————————-
Task: {stringified task inputs and outputs}
LARC Description: {description of the task-1 from LARC dataset}
Good Description: {hierarchical description}
————————-
*[truncated]*
————————-
Task: {stringified task inputs and outputs for task-K}
LARC Description: {description of the task-K from LARC dataset}
Good Description: {hierarchical description}
————————-
Task: {stringified task inputs and outputs for query task}
LARC Description: {description of the query task from LARC dataset}

---

## D.2. Few-shot Prompting Details

We use the following simple prompting template with k-shot prompting for all data generation procedures, where numbers filled with examples sampled from seed set. In simple few-shot generation, we exclude examples. We use GPT-4 and GPT-4o to generate the new scripts.

---

You are a problem generator on 2D grids of colors. Here are some examples of such transformations, please follow the format:
————————-
Example: {description of the generator function-1}
Script: {generator function-1}
————————-
[truncated]
————————-
Example: {description of the generator function-K}
Script: {generator function-K}

Please generate more and make sure they are different:

---

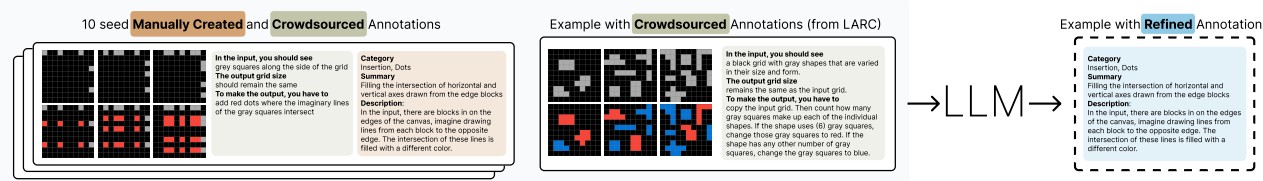

*Figure 15.* **Generating quality seed descriptions:** We use few-shot prompting to generate descriptions for a given task, using 10 manually created seed descriptions along with crowd-worker annotations from Acquaviva et al. (2022) as few-shot examples. For a given new task, we similarly provide the LM with examples and crowd-worker annotations (available only for training tasks).

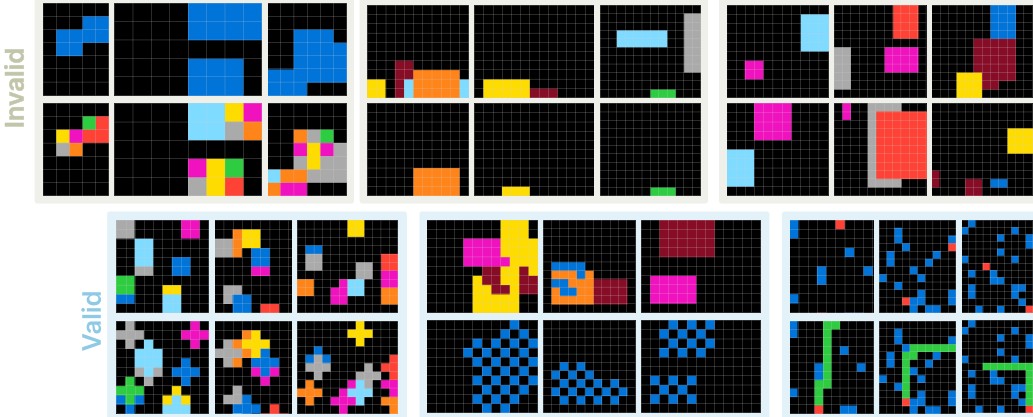

*Figure 16.* **Example tasks generated by LM data augmentation procedure:** We display three reasonable tasks that we can infer a simple transformation (valid), and three tasks that we could not infer a simple transformation (invalid).

# E. Augmented Inference Pipeline

## E.1. Augmented Inference

Recent work has shown that scaling test-time compute can significantly improve the performance of LMs. One of the most common techniques to do this is by sampling multiple responses, and then selecting the best response using a ranker. However, while sampling is very effective in domains with multiple possible solutions (programs in code) or multiple possible paths to the final answer (math), it can be detrimental when generating answers directly, as there is no way to directly enforce diversity *across* samples while ensuring coherence *within* samples. As an alternative inference-time scaling, we use an *augmented inference* strategy that generates multiple prediction candidates by using geometric transformations, combined with a greedy decoding scheme.

For a given task with training examples $(x_k, y_k)_{k=1}^{K}$ and test input $x_{\text{test}}$, we use invertible geometric transformations to produce equivalent transformed versions of the task, as shown in Figure 5. Let $\mathcal{T}$ be some set set of invertible geometric transformations (e.g., rotations and reflections). For each transformation $t \in \mathcal{T}$, we apply $t$ to all training demonstrations and the test input and run our model with these transformed inputs. We then apply the inverse transformation to obtain the final prediction for that transformation.

$$\tilde{y} \sim \text{LM}(t(\mathbf{d}_{\text{input}})) := [t(x_1), t(y_1), \ldots, t(x_{\text{test}})] \tag{6}$$

$$y_t = t^{-1}(\tilde{y}) \tag{7}$$

We further augment our predictions by permuting the order of training examples. For each transformation $g$, we sample $n = 2$ different permutations of the demonstration sequence, resulting in $n \cdot |\mathcal{T}|$ total predictions per task. This is to mitigate any bias in the model's processing of the demonstration sequence. (Bober-Irizar & Banerjee, 2024) also find transpose and rotation is helpful to produce extra prediction candidates.

## E.2. Ensembling Predictions (Voting Strategy)

We employ a hierarchical voting strategy to determine the final prediction from the set of candidates $\{y\}_{i=1}^{n \cdot |\mathcal{T}|}$. This approach involves two stages of voting to progressively narrow down the best candidates: first, by selecting the most frequent predictions within each transformation, and then by conducting an overall vote across transformation-specific candidates to identify the top-2 most frequent predictions. The details of each stage are as follows:

1. **Intra Transformation Voting:** We group predictions by their corresponding transformation $t$ and select the top-3 most frequent predictions within each group. If fewer than 3 unique predictions exist within a group, we supplement the candidates by computing additional predictions through:

   - **Row-based majority**: For each row in the predicted output grid, we take the most frequent row values across all predictions in the transformation group.
   - **Column-based majority**: Similarly, for each column in the predicted output grid, we take the most frequent column values across all predictions in the transformation group.

2. **Global Voting:** Using the selected transformation-specific candidates obtained from (1), we conduct an overall vote to select the top-2 most frequent predictions for submission. In case of a tie, predictions with the identity transformation are given priority.

# F. BIG-Bench Hard Details

## F.1. Further Experimental Details

We write our own evaluation function for BIG-Bench Hard available in our codebase. We found that existing evaluation frameworks did not properly measure zero-shot performance due to insufficient answer-extraction parsing and answer-format prompting. We also wanted more control in splitting each individual task's dataset into demonstration examples and evaluation sets. For all results, we average results over different selections of the 10 few-shot examples with the following 5 random seeds: $42, 43, 44, 45, 46$. The full TTT and inference process takes approximately 15 minutes on an NVIDIA A100 GPU.

The standard error of the mean for each method in Figure 8 over the 5 seeds is given in Table 7.

*Table 7.* Standard Error of the Mean for each method in Figure 8.

| Method | Standard Error of the Mean |
|---|---|
| Zero-Shot | 0.01 |
| ICL | 0.19 |
| TTT | 0.20 |
| No Example Permutation | 0.32 |
| E2E | 0.66 |
| Shared TTT | 0.72 |
| No Demo Loss | 0.69 |
| Loss on Inputs and Outputs | 0.35 |

We search over the following hyperparameters:

*Table 8.* BBH TTT Fine-tuning Hyperparameters

| Hyperparameter | Search Space |
|---|---|
| learning rate | [1e-5, 5e-5, **1e-4**, 3e-4] |
| $r$ LoRA rank | [**64**, 128] |
| $\alpha$ LoRA alpha | [16, 32, **64**, 128] |
| epochs | 1 |
| batch size | 5 |
| training steps | [20, **40**, 60] |
| optimizer | AdamW |
| scheduler | Cosine LR Schedule |

We similarly use the torchtune(torchtune Maintainers & Contributors, 2024) library for test-time training and the vLLM (Kwon et al., 2023) library for inference.

## F.2. Task-specific Results

The full results for all tasks over all methods and ablations are shown in Figure 17.

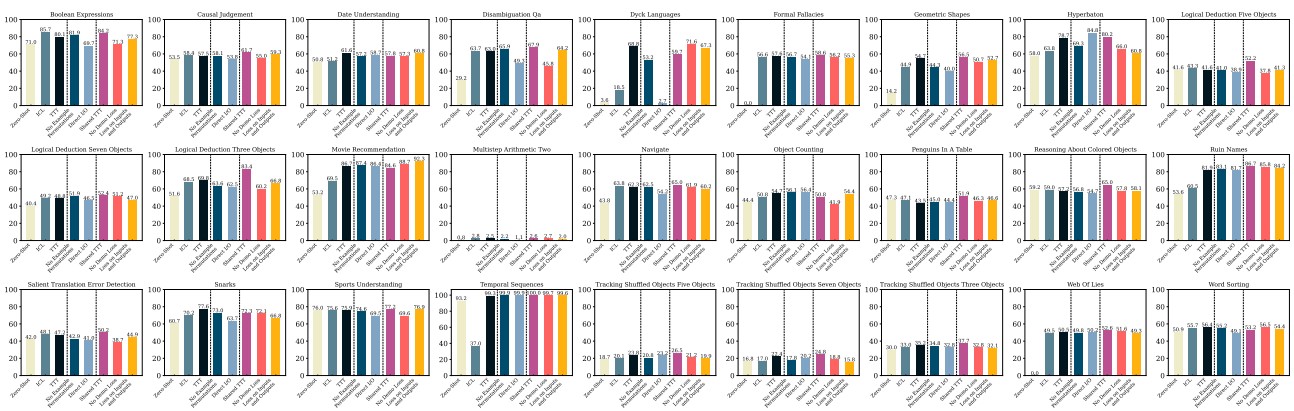

*Figure 17.* Task-specific 10-shot results for each BIG-Bench Hard task, averaged over 5 random seeds.

| **Range for** $x =$ TTT accuracy $-$ ICL Accuracy | **Tasks (Count)** |
|---|---|
| $x \leq -5$ | 1 task total: *Boolean Expressions* |
| $-5 < x \leq -2$ | 1 task total: *Penguins In A Table* |
| $-2 < x < 2$ | 13 tasks total: *Causal Judgement*, *Disambiguation Qa*, *Formal Fallacies*, *Logical Deduction Five Objects*, *Logical Deduction Seven Objects*, *Logical Deduction Three Objects*, *Multistep Arithmetic Two*, *Navigate*, *Reasoning About Colored Objects*, *Salient Translation Error Detection*, *Sports Understanding*, *Web Of Lies*, *Word Sorting* |
| $2 \leq x < 5$ | 3 tasks total: *Object Counting*, *Tracking Shuffled Objects Five Objects*, *Tracking Shuffled Objects Three Objects* |
| $x \geq 5$ | 9 tasks total: *Date Understanding*, *Dyck Languages*, *Geometric Shapes*, *Hyperbaton*, *Movie Recommendation*, *Ruin Names*, *Snarks*, *Temporal Sequences*, *Tracking Shuffled Objects Seven Objects* |

*Table 9.* Tasks Categorized by the Difference $x =$ TTT Accuracy $-$ ICL Accuracy.

