# OpenReview forum: "The Surprising Effectiveness of Test-Time Training for Few-Shot Learning"
_ICML.cc/2025/Conference — ICML 2025 poster_

### Official Review · Reviewer_fvyu · 2025-03-14

**Overall Recommendation:** 1

**Summary:**

This paper presents a comprehensive analysis of the Abstraction and Reasoning Corpus (ARC) and BIG-Bench Hard (BBH) tasks. The authors train LoRA as an adapter. For training, the authors use flips, rotations, and color permutations to augment the original training data. During inference, they adopt intra-transformation voting and global voting strategies. Finally, they achieve improvements of 28% and 7% on ARC and BBH, respectively.

## Update
Thanks for authors' detailed response. I appreciate authors re-state their contribution more clearly. This paper primarily focus on data augmentation and in-context fine-tuning. I believe that with further explanation and clarification, this paper will be much clearer. However, as for the submitted version, there are some missing important points, including augmented data statics, lora baselines comparison, more clearly statement of contribution, and so on (see above).
I will maintain my score. I think this paper is not suitable for publication at this time

**Claims And Evidence:**

There are some over-claims or unclear definitions here that need further clarification:
1. Test-time training: creating a self-supervised learning problem based on this single test samplex, updating θ at test time before making a prediction (quoted from [1]). This paper adopt a lora to train on trianing data, then using several inference scaling methods to test. **The defintion of TTT is different with the main method in this paper.**
2. The Lora is a parameter-efficient fine-tuning methods. Authors use this method as baselines. However, I believe more details should be provided regarding the inference methods used in the baseline approach. Most of the improvements come from the data augmentation and voting strategies presented in the paper.
3. **It should be clearly stated in contribution that this paper employed existing techniques to conduct a comprehensive analysis on ARC and BBH, including data augmentation and voting strategies.  This paper does not propose novel methods, theoretical results, or insights.**
---
[1] Sun Y, Wang X, Liu Z, et al. Test-time training with self-supervision for generalization under distribution shifts[C]//International conference on machine learning. PMLR, 2020: 9229-9248.

**Essential References Not Discussed:**

This published paper proposed training a lora during the inference time, which is related to this paper.
[2] Wang Y, Ma D, Cai D. With greater text comes greater necessity: Inference-time training helps long text generation[J]. arXiv preprint arXiv:2401.11504, 2024.

**Experimental Designs Or Analyses:**

Their empirical analysis is reasonable and comprehensive. My main concern is that the baseline should be compared using the same inference methods.

**Methods And Evaluation Criteria:**

This paper doesn't propose new methods. However, their empirical analysis is thorough and demonstrates significant effectiveness. Their evaluation criteria is reasonable.

**Other Comments Or Suggestions:**

The formulation differences between LoRA and test-time training (TTT) should be explicitly clarified.

**Other Strengths And Weaknesses:**

More theoretical analysis is necessary for out-of-distribution tasks, especially for the ARC and BBH tasks studied in this paper.

**Questions For Authors:**

Please provide dataset statistics for LoRA training, such as the dataset sizes obtained after each type of augmentation, to better analyze the sources of actual performance improvements.

**Relation To Broader Scientific Literature:**

This paper is not related to test-time training. They use LoRA, a parameter-efficient fine-tuning method, rather than conducting training during inference.

**Theoretical Claims:**

This paper does not provide theoretical results for insights.

---

> ### Author Rebuttal · Authors · 2025-04-01
>
> Thank you for your review and the feedback on our paper.
>
> **Q1. Definition and Application of Test-Time Training (TTT)**
>
> We use the definition of TTT provided by ([Sun et al. (2020)](https://arxiv.org/abs/1909.13231)): self-supervised training a model using the unlabeled test sample $d_{\textrm{input}}$. Our work applies this definition in the few-shot learning setting where $d_{\textrm{input}}$ consists of instance-specific few-shot demonstrations and a query. The core idea remains: adapting model parameters at inference time based on information available for the specific test input being processed.
>
> Finally, LoRA is simply the means by which we implement TTT in this paper. it is a simple and efficient fine-tuning method that enables us to do TTT with a 8B-size model. In practice, our TTT framework can be used with other fine-tuning techniques as well.
>
> We can clarify more if the reviewer has more questions.
>
>
> **Q2. Most of the improvements come from inference strategies**
>
> We kindly disagree. Our experiments on the Big-Bench Hard (BBH) show that the TTT without any task-specific voting mechanisms or data augmentations still yields significant improvements. Moreover, our vanilla method (Fig. 7), which **excludes any augmentations or voting procedures**, still achieves a substantial performance boost—solving 22 tasks compared to just 5 (no TTT), marking a 440% improvement.
>
> **Q3. Can you clarify the contributions and significance?**
>
> Thank you for asking for clarification, and we will update the text to reflect our contributions.
>
> We presented a successful extension of the TTT paradigm to the few-shot learning setting and did a systematic study of design choices for this setting. To the best of our knowledge, we are the first to apply TTT within the in-context learning framework.
>
> We showed substantial improvements (e.g., up to 6x on ARC, 7.3pp on BBH) over strong baselines on 2 popular benchmarks. The magnitude of these improvements highlight the limitations of standard ICL and opens up a new avenue for LM research with TTT.
>
> **Q4. TTT Dataset Statistics**
>
> The size of the dynamically generated $D_\textrm{TTT}$ depends on the number of demonstrations (K) and the number of augmentations/permutations used.
>
> For ARC, we use $K$ demonstrations (typically 2-7), generate $K$ leave-one-out tasks, apply $|T|$ geometric transformations (Section C.1 lists 20 base transformations/compositions), and n=2 permutations (line 1066), leading to roughly $K * |T| * n$ synthetic examples per task, capped at 250 (line 202). For BBH, we use $K=10$ demonstrations and $n=40$ permutations (line 344), resulting in $10 * 40 = 400$ leave-one-out examples per task.
>
> We will add these specific calculations/details to Appendix C.2 (for ARC) and F.1 (for BBH) to clarify the scale of the temporary training data used for TTT.

---

> > ### Comment · Reviewer_fvyu · 2025-04-03
> >
> > Thank authors for kindly response. I think there are still three points authors should clarify :
> >
> >  (1) Did author use labels in "test time training" ?
> >
> > Figure 2 indicated that they use the outputs' loss on augmented data. However, the common test time training methods are training test set without labels.
> >
> >  (2)  How many data used in "test time training" ?
> >
> > Lets ref to recent test time training papers, these work demonstrate test time training is learning during the inference. And the training sample is very scare. However, this paper use the data augmentation as main method as listed in 3.1. And the statics of augmentated data is missing in the submission (authors provide partial information in rebuttal).
> >
> > (3) there is already Lora-style test time training.
> >
> > I have already mentioned this paper in my first version of review. Considering this paper using the same method, I think authors should provide a baseline experiments to give credit to similar published paper.
> >
> > [1] With Greater Text Comes Greater Necessity: Inference-Time Training Helps Long Text Generation
> >
> > ----
> > In conclusion, this paper proposed a data augmentation and Lora fine-tuning pipeline for ARC tasks. To the best of my knowledge, all this methods are well-studied, which raised my concerns about this paper's contribution to the commuity. Additionally, this paper also presents no new theoretical results for ICML submissions.
> >
> > I will keep my score 1 "reject".

---

> > > ### Author Response · Authors · 2025-04-05
> > >
> > > **(1) “Did author use labels in "test time training"?”**
> > >
> > > We use demonstration labels, not the test labels. This paper is about TTT+ICL in the few-shot setting, so few-shot labels will of course be used.
> > >
> > > Note that each ARC task (or instance) **is defined by** the given 2-7 demonstrations — they are collectively **the input** in an ARC task. For example, in Figure 2,$(x_1, y_1), (x_2, y_2), (x_3, y_3), (x_4, )$ is the complete input of the task, and $y_4$ is the output. Each $(x_i, y_i)$ pair within this sequence is a single demonstration. The objective for an ARC task is to deduce/learn the underlying transformation rule from the demonstrations and then apply this rule to the final query input ($x_4$) to predict the corresponding query output ($y_4$). Only $y_4$ is **the test label** for the ARC task instance, and it is not accessed or used during our TTT process.
> > >
> > > The same goes for the BB-Hard benchmark, comprising 27 distinct tasks. While BB-Hard tasks differ in that they can be solved without demonstrations, providing few-shot examples significantly enhances performance, a standard few-shot learning scenario used in previous research. Treating these few-shot examples as a mini-training set at test time (which we refer to as the 'Direct I/O' baseline) also improves performance. Our TTT method builds upon and combines these ideas, achieving the most substantial performance gains, as evidenced in Figure 8. Again, the actual test output for a BB-Hard query is never used during TTT.
> > >
> > > In Figure 2, we show our methodology of constructing *synthetic* tasks based on the given demonstrations given in the test input. We believe this directly falls under the definition of test-time training as we’re updating model parameters based on the test inputs before making a prediction [1].
> > >
> > > **(2) “How many data used in "test time training"?”**
> > >
> > > We would like to emphasize that our experiments on BB-Hard (Section 5) and ablations on ARC (Figures 5, 7) show that task-specific augmentations are not necessary to achieve major improvements with our method.
> > >
> > > Previous work on test-time training [1] also makes use of data-augmentations, and we believe this is quite useful in expanding the TTT dataset and improving generalization.
> > >
> > > We provide in-depth details of the data augmentations used on ARC and their applications in augmented inference in appendices C and E, and will include the stats we reported in the rebuttal to the revision. Please let us know if there are any further details you believe we should provide.
> > >
> > > **(3) “there is already Lora-style test time training.”**
> > >
> > > Thank you for pointing out Wang et al. (2024) paper. This is indeed relevant work showing inference-time LoRA updates. We plan on citing this paper and discussing its connection and distinction. It focuses on long text generation whereas our focus is on few-shot learning and reasoning tasks.
> > >
> > > We believe it’s incorrect to say that all methods we used are well-studied. While individual components like TTT or LoRA updates at test-time are known, the main idea of the paper is to combine TTT with ICL, which is fundamentally novel. To the best of our knowledge, the *combination* and *systematic study* of Test-Time Training specifically within the In-Context Learning setting (TTT+ICL) has not previously been explored. The improvements over standard ICL with this method is very significant, and we believe it will be very interesting to many audience at ICML 2025.
> > >
> > > **(4) “No new theoretical results for ICML submissions.”**
> > >
> > > Our paper is not a theory paper! This paper presents a careful empirical analysis and significant improvements to few-shot learning abilities of language models. Please kindly refer to ICML 2025 Call For Papers document for different categories of acceptable papers.
> > >
> > > [1] Sun et al. Test-Time Training with Self-Supervision for Generalization under Distribution Shifts. 2019.

---

### Official Review · Reviewer_LZSp · 2025-03-14

**Overall Recommendation:** 4

**Summary:**

This paper investigates test-time training (TTT) for improving language models' few-shot learning capabilities, particularly on novel tasks that require reasoning and abstraction.

- TTT significantly improves performance on challenging reasoning tasks, e.g. on ARC-AGI, TTT with in-context examples yields up to 6× higher accuracy (53.0%) compared to fine-tuned baselines, and reaches 61.9% when ensembled with program synthesis methods.
- On BIG-Bench Hard (BBH), TTT improves performance by 7.3 percentage points over standard few-shot prompting (50.5% to 57.8%).


## Update after rebuttal

I maintain my rating as accept.

**Claims And Evidence:**

Nothing to report.

**Essential References Not Discussed:**

None

**Experimental Designs Or Analyses:**

Nothing to report.

**Methods And Evaluation Criteria:**

Nothing to report.

**Other Comments Or Suggestions:**

None

**Other Strengths And Weaknesses:**

None

**Questions For Authors:**

None

**Relation To Broader Scientific Literature:**

Nothing to report.

**Theoretical Claims:**

Nothing to report.

---

> ### Author Rebuttal · Authors · 2025-04-01
>
> Thank you for the positive review! We are happy to answer any new questions you have later during the rebuttal process.

---

### Official Review · Reviewer_nbYh · 2025-03-19

**Overall Recommendation:** 3

**Summary:**

This paper proposes to use test-time-training as a method for scaling test-time capabilities of large models under the few-shot setting, and tested the model's performance on ARC and BBH bechmarks. The experiment results show positive performance.

**Claims And Evidence:**

Main claim: TTT helps few-shot performance compared to In Context Learning. The claim is very well supported by various ablation studies in BBH and ARC.

**Essential References Not Discussed:**

No key missing references

**Experimental Designs Or Analyses:**

The authors clearly discussed and ablated the model components.

**Methods And Evaluation Criteria:**

Methods

(1) Three methods for TTT: leave-one-out TTT, direct train TTT, augmentated data for TTT. These three are ablated in the experiments

(2) For training loss: test loss + all output loss + per-token loss.

Evaluation: ablation on different methods, losses, and metrics come from the benchmark. So it's sound and solid.

**Other Comments Or Suggestions:**

The paper did use many data-aug or voting mechanisms to make it work better. It'll be cleaner and cooler if there's no need for those.

**Other Strengths And Weaknesses:**

Very well written paper.

**Questions For Authors:**

I wonder whether authors have tried other domains such as coding or algebraic / math questions?

Recent findings show RL-trained models generalizes better, do the authors think TTT can still hold its advantage if the base model is strongly RL-trained, or the baseline is not FT but actually RL finetuned?

Also, many benefits disappear when actual scaling happens, not sure what will happen for this TTT finding.

(I think the paper's finding is interesting overall. Just curious about the authors' thoughts on the above questions.)

**Relation To Broader Scientific Literature:**

This would be interesting results to the "general reasoning" audience.

**Theoretical Claims:**

No theoretical claim.

---

> ### Author Rebuttal · Authors · 2025-04-01
>
> **Q1. Have authors tried domains such as coding or math?**
> This paper produced a SoTA way to apply TTT to LMs in the few-shot learning setting using the challenging ARC-AGI and BB-Hard as our reasoning problem sets. Note that our implementations do not leverage the CoT capabilities of the models. We believe extending this to non-few-shot learning settings and to domains where CoT is crucial is a very exciting research problem and we’re currently exploring these extensions.
>
> **Q2. Recent findings show RL-trained models generalize better. Can TTT still be advantageous if the base model is strongly RL-trained?**
> That is a great question! Recently, there has been a lot of interest in using RL to unlock long reasoning abilities in LMs. As mentioned in our previous answer, combining TTT with CoT models (both RL or no RL)  is a promising future direction! Similarly, another possible extension is **Test-Time-RL**, where RL is incorporated into the test-time training process ([Simonds and Yoshiyama](https://arxiv.org/abs/2503.00735), 2025).
>
> **Q3. Many benefits disappear when actual scaling happens, how will this affect TTT?**
> In Section 4.4, we present scaling results for Llama models of 1B, 3B, and 8B parameters, where TTT improves performance by 480%, 163% and 157% respectively. Thus, TTT scales effectively with increasing model size.
>
>
> **Q4. Augmentations makes the method a little complicated**
> Our experiments on the Big-Bench Hard (BBH) suite of tasks show that the core method of fewshot TTT without any task-specific voting mechanisms or data augmentations still yields significant improvements. Moreover, our vanilla method (Fig. 7), which excludes any augmentations or voting procedures, still achieves a substantial performance boost—solving 22 tasks compared to just 5 (no TTT), marking a 440% improvement.

---

### Decision · Program_Chairs · 2025-05-01

**Decision:**

Accept (poster)

**Comment:**

The paper proposes using test-time training (TTT) to improve LLM reasoning and few-shot learning on tasks that require reasoning or rule-based generalization. The authors demonstrate that TTT, which involves updating model parameters during inference, enhances performance on the ARC and BBH benchmarks.

The paper shows strong empirical results, with TTT improving fine-tuned model accuracy on ARC by approximately 6x and increasing accuracy on BBH by 7.3 percentage points.  The paper is well-written and the claims are generally well-supported by the evidence provided.

One reviewer argues that the paper does not propose novel methods, theoretical results, or insights, and that the improvements mainly come from data augmentation and voting strategies. The authors replied that they are the first to apply TTT within the in-context learning framework, and that the combination and systematic study of TTT in this setting is novel.

Overall, the paper demonstrates the effectiveness of test-time training in improving few-shot learning capabilities. The authors addressed reviewer concerns in their rebuttal. Hopefully, the authors can address Reviewer fvyu 's comments on "augmented data statics, lora baselines comparison, more clearly statement of contribution".

Overall, we recommend acceptance.